# Automatic Detection of Changes in Signal Strength Characteristics in a Wi-Fi Network for an Indoor Localisation System

**DOI:** 10.3390/s20071828

**Published:** 2020-03-25

**Authors:** Marcin Luckner, Rafał Górak

**Affiliations:** Faculty of Mathematics and Information Science, Warsaw University of Technology, Koszykowa 75 street, 00-662 Warsaw, Poland

**Keywords:** Quality of Service, system deployment and maintenance, Wi-Fi network, indoor localisation system, fingerprinting

## Abstract

This paper faces the issue of changing the received signal strength (RSS) from an observed access point (AP). Such a change can reduce the Quality of Service (QoS) of a Wi-Fi-based Indoor Localisation System. We have proposed a dynamic system based on an estimator of RSS using the readings from other APs. Using an optimal threshold, the algorithm recognises an AP that has changed its characteristics. Next, the system rebuilds the localisation model excluding the changed AP to keep QoS. For the tests, we simulated a change in the analysed Wi-Fi network by replacing the measured RSS by an RSS obtained from the same AP model that lies in another place inside the same multi-floor building. The algorithm was evaluated in simulations of an isolated single-floor building, a single-floor building and a multi-floor building. The mean increase of the localisation error obtained by the system varies from 0.25 to 0.61 m after the RSS changes, whereas the error increase without using the system is between 1.21 and 1.98 m. The system can be applied to any service based on a Wi-Fi network for various kinds of changes like a reconfiguration of the network, a local malfunction or ageing of the infrastructure.

## 1. Introduction

Localisation services–including indoor ones–are becoming more and more common. Because of the business application of indoor localisation systems, the issue of providing an accurate indoor position becomes critical and requires a specific level of accuracy from the localisation system. The most popular indoor localisation system type is based on a vector of received signal strength (RSS) calculated for access points (APs) from internal Wi-Fi infrastructure. The advantages of the system are low cost and accessibility, as the measurements can be performed on almost every mobile device with a Wi-Fi module.

Very often, such a system is trained using fingerprinting. In the localisation process, the current position can be determined by comparing the RSS vector to a created map of fingerprints. Different environments, test areas, and sensor technology have a significant impact on the Quality of Service (QoS)–measured by the localisation error–of the indoor localisation system [1]. However, in the case of a localisation system based on fingerprinting there is one important additional factor: The QoS can drop if there are some changes in the characteristics of the RSS that were used to train the system.

There are various reasons for changes in system characteristics. One of them regards the model readings from a mobile AP during the creation of learning set. A mobile AP can change its position. As a result, the broadcast signal can be significantly different or even not observed. In performed experiments, Leca et al. [2,3] have estimated that approximately 3–4% of APs observed in an outdoor environment are mobile APs. Their existence in the learning data can increase the log-Gaussian Mean Error even by 37%, although the influence of mobile APs is not explicit for other error measures. Another reason for the changes is a modification of the Wi-Fi infrastructure. According to current needs, the network can be reconfigured to optimise the trade-off between the range and the latency [4]. The last reason is the ageing of AP devices. Górak et al. [5] have shown that the QoS of the localisation system based on unmodified Wi-Fi infrastructure deceases from 4 to 5 m over two years.

In this work, we propose a detection algorithm that can detect changes in AP signal characteristics. Using the Random Forest algorithm, we create an estimator that predicts the signal strength of the AP, based on the readings from all the remaining APs. Next, for every AP, we calculate an optimal threshold that will allow us to compare a predicted signal strength to the actual strength. If the difference between the prediction and the recorded signal strength is above the threshold, we predict that the AP has changed its characteristics for reasons other than typical environmental factors and this may substantially influence the accuracy of the localisation. If this is the case, we rebuild the localisation model, by excluding this AP from the learning data.

To evaluate our algorithm, we created a unique testbed. The change is simulated by replacing AP signals by signals of an AP from a different location. Both APs are the same model (Cisco AIR-LAP1142N-E-K9). Therefore, it is practically the same as physically moving one AP to a different location. Moreover, our approach allows us to avoid modelling of the propagation of signal strength in the dynamic environment of a public building, which is a very difficult task. Instead, we gather a huge amount of fingerprinting data (millions of records), each record containing many features (RSS readings from many APs). Next, we use machine learning methods to detect APs with substantially changed characteristics. After the detection of changes, the algorithm starts the recalculation of the localisation model.

Our algorithm was tested inside the big modern multi-floor building of the Faculty of Mathematics and Information Science (MIS) of the Warsaw University of Technology. The algorithm automatically detected simulated changes in the Wi-Fi infrastructure and triggered the recalculation of the localisation model. As a result, our solution reduces localisation errors created by changes.

The remaining part of the paper is organised as follows. In Section 2, the related work is discussed. Section 3 presents the proposed localisation model and describes how the system detects the changes. The created simulations and data sets of fingerprints collected in the MIS building are described and discussed in Section 4. The results obtained for three scenarios are described in Section 5. Section 6 presents our conclusions.

## 2. State-of-the-Art

Tuta et al. [6] stressed the issue of developing an indoor positioning system with the main aim of making it useful for real-world deployments, including the creation of self-calibrating and self-adaptive systems. Several such localisation systems were proposed.

Cai et al. [7] proposed an adaptive indoor localisation system—an integration of received signal strength indication (RSSI)-based and inertial navigation system (INS)-based approaches—called coupled RSSI and INS localisation (CRIL). The system used the results from RSS and INS and updated the channel model in the RSSI in real-time. However, the adaptation is based on infrastructure of anchors of known location. Tuta et al. [6] merged a free-space path loss model and a propagation model to create a self-calibrating and self-adaptive model. This procedure infers parametres of the space and simulates the propagation of the signal. The historical points are used for localisation improvement.

Several solutions use a specialised chipset. Nevat et al. [8] proposed two-way time-of-arrival ranging devices to perform localisation. The approach was based on nonlinear regression analysis, where the missing observations were treated as Missing Not at Random. A similar idea was proposed by Batstone et al. [9,10]. However, such solutions are hard for broad implementation on commonly used mobile phones [11].

Several works aim to eliminate the missing signals. Lin et al. [12] recovered a fingerprinting map created during the fingerprinting process in place of the missing signals. Chang et al. [13] proposed a similar solution. Saleem et al. [14] discussed—on a laboratory testbed—how to recover missing APs’ RSS if the radio map covers all measurement points for all APs. Górak et al. [15] detected the missing signals and eliminated their sources from the localisation model.

Designing our solution, we assumed that the system should work on commonly used devices. Therefore, the solution cannot be based on a specialised chipset as in [8,9,10]. Our solution can work using standard Wi-Fi APs and mobile devices. Also, the designed system should work with unreliable infrastructure. Therefore, the concept of Wi-Fi anchors used in [7] cannot be applied. The anchors can be a major source of system weakness in the case of changes in their RSS characteristics. The central concept of our solution lies in the detection of such changes. The main works that analyse the localisation QoS are focused on managing lack of data [12,13,14,15]. Although the detection of missing signals is critical for QoS, we extend this idea to detect all significant changes in the observed signals. Finally, to design a universal solution, we avoid signal propagation modelling [6] and the necessity of a knowledge of the Wi-Fi infrastructure scheme [7] using by fingerprinting and machine learning techniques.

The method proposed in this work could be extended using crowdsourcing [16], as we did previously to detect a disabled AP [15]. The previously introduced methods allow the system to limit rebuilding frequency by waiting for a significant number of raised alarms to recreate the localisation model. Moreover, the proposed methods can extend a set of observed APs. However, the current work introduces the new concept of RSS characteristics changes detection and we intentionally limited additional factors during the experiments. This aspect could be an area of future work.

The tests in our work were performed on data collected in the multi-floor MIS building. Other analyses for the building—multi-floor localisation and feature selection—can be found in [15,17,18,19,20]. Mostly, we cannot compare our results directly with the mentioned works. However, we performed a comparison of the proposed system with the approaches presented in [14,15].

## 3. Methodology

### 3.1. Localisation Model

Following the work in [15], let us formally define the localisation problem which we deal with in the following part of this work.

By $S_{L}^{\mathrm{AP}}$ and $S_{T}^{\mathrm{AP}}$ we denote the learning and testing data sets, respectively. They consist of vectors (fingerprints) $\left(r_{1},r_{2},\ldots ,r_{n},t,x,y,z\right)$, where $r_{p}$ is the RSS reading from the *p*th AP, from a given set of APs AP, at the time *t*, in the location (x,y,z) where x,y are the horizontal coordinates, and *z* is the vertical one.

For a subset ${\mathrm{AP}}^{\prime }\subset \mathrm{AP}$, we consider new data sets $S_{L}^{{\mathrm{AP}}^{\prime }}$, $S_{T}^{{\mathrm{AP}}^{\prime }}$ that are modified data sets $S_{L}^{\mathrm{AP}}$, $S_{T}^{\mathrm{AP}}$, respectively:(1)$$\begin{array}{ccc} S_{L}^{{\mathrm{AP}}^{\prime }} & = & \{\left({\left(r_{i}\right)}_{i\in {\mathrm{AP}}^{\prime }},t,x,y,z\right):\;\left({\left(r_{i}\right)}_{i\in \mathrm{AP}},t,x,y,z\right)\in S_{L}^{\mathrm{AP}}\}; \end{array}$$(2)$$\begin{array}{ccc} S_{T}^{{\mathrm{AP}}^{\prime }} & = & \left\{\left({\left(r_{i}\right)}_{i\in {\mathrm{AP}}^{\prime }},t,x,y,z\right):\;\left({\left(r_{i}\right)}_{i\in \mathrm{AP}},t,x,y,z\right)\in S_{T}^{\mathrm{AP}}\right\} \end{array}$$

In other words, we take in $S_{L}^{\mathrm{AP}}$, $S_{T}^{\mathrm{AP}}$ only RSS readings from APs from ${\mathrm{AP}}^{\prime }$.

**Problem** **1.***Construct a localisation model based on a measurement series $S_{L}^{\mathrm{AP}}$ (learning series). The localisation model is a function $\hat{L}:R^{n}\mapsto R^{2}$ (2D case, z is constant) or $\hat{L}:R^{n}\mapsto R^{3}$ (3D case), depending on the localisation area. For a given RSS vector $r=\left(r_{1},r_{2},\ldots ,r_{n}\right)\in R^{n}$, $\hat{L}\left(r\right)$ estimates the location where the measurement v was taken*.

The localisation issue formulated as Problem 1 is not a tracking problem and the historical RSS readings are not taken into account. The localisation is based on a single RSS reading from multiple APs.

To evaluate the model, we introduce the following standard measures of QoS.

**Definition** **1.**
*Let $S_{T}^{\mathrm{AP}}$ (testing series) be a measurement series and $\hat{L}$ a localisation model. For an element $s=\left(r_{1},r_{2},\ldots ,r_{n},t,x,y,z\right)\in S_{T}$, we introduce the the following natural error measurement,*
(3)$$E(\hat{L},s)=\sqrt{{\left(\hat{x}-x\right)}^{2}+{\left(\hat{y}-y\right)}^{2}+{\left(\hat{z}-z\right)}^{2}}$$
*where $\hat{L}\left(r\right)=\left(\hat{x},\hat{y},\hat{z}\right)$. In other words, although (x,y,z) is the true position of fingerprint s, $\left(\hat{x},\hat{y},\hat{z}\right)$ is its predicted position based on RSS vector r.*


**Definition** **2.**
*For a testing series $S_{T}$ and the localisation model $\hat{L}$, let us define the following QoS measures,*
(4)$$\begin{array}{cc} \mathrm{Mean}\mathrm{error}: & \mathrm{mean}\{E(\hat{L},s):\;s\in S_{T}\}; \end{array}$$
(5)$$\begin{array}{cc} \mathrm{Median}\mathrm{error}: & \mathrm{median}\{E(\hat{L},s):\;s\in S_{T}\}; \end{array}$$
(6)$$\begin{array}{cc} \mathrm{Gross}\mathrm{error}: & {\mathrm{perc}}_{80}\left\{E(\hat{L},s):\;s\in S_{T}\right\}. \end{array}$$


The goal of localisation is minimising these measures.

We construct the localisation model $\hat{L}:R^{n}\mapsto R^{3}$ using the Random Forest algorithm as presented in [21]. This will be one of the main parts of the localisation solution. The main advantages of the algorithm are speed and high quality. Alternatively, the AdaBoost algorithm can be used [22].

First, we create estimators ${\hat{L}}_{x}$, ${\hat{L}}_{y}$ and ${\hat{L}}_{z}$ by applying the Random Forest algorithm where the training set is $S_{L}$, which predicts separately coordinates of *x*, *y* and *z*, respectively. For creating the estimators, regression trees are grown. The selected number of grown trees is 100 as the analysis based on $S_{L}$ (see, e.g., in [15]) suggests that growing more trees does not improve QoS.

### 3.2. Detection of Changes of AP Location

Let us describe the method of detecting that the RSS characteristics of AP *p* have changed. The method presented in Figure 1 starts denoting for vector $r\in R^{n}$ a vector ${\overset{\vee }{r}}_{p}=\left(r_{1},r_{2},\ldots ,r_{p-1},r_{p+1},\ldots \right)={\left(r_{i}\right)}_{i\neq p}\in R^{n-1}$, which is vector r with the *p*th coordinate removed. For a learning data set $S_{L}^{\mathrm{AP}}$ we define a set of such vectors ${\mathrm{RSS}}_{p}=\left\{{\overset{\vee }{r}}_{p}\;|\;\exists \left(t,x,y,z\right):\left(r,t,x,y,z\right)\in S_{L}^{\mathrm{AP}}\right\}$. Therefore, we have a function (possibly a multifunction) $g_{p}$ that for a given vector ${\overset{\vee }{r}}_{p}\in {\mathrm{RSS}}_{p}$ returns a missing coordinate $r_{p}\in R$ of RSS from the removed AP. Now, based on $g_{p}$ and using the Random Forrest algorithm once again, we create an estimator, ${\hat{g}}_{p}:R^{n-1}\mapsto R$, that predicts the RSS of the removed AP *p*, based on the readings from all the remaining APs.

We define the estimation error of the RSS prediction as $|{\hat{g}}_{p}\left({\overset{\vee }{r}}_{p}\right)-r_{p}|$, where $\left(r,t,x,y,z\right)\in S_{L}^{\mathrm{AP}}$ and $r_{p}$ is the *p*th coordinate of r. Next, we calculate the threshold *t* that separates errors obtained for valid characteristics from the rest. Regarding typical noises that can occur in a real environment, the estimation errors for valid characteristics will be higher than zero. However, we assume that errors introduced by a change (such as a change of AP location) will be significantly higher than typical errors. If not, we can assume that the influence of the change on QoS will not be substantial.

The threshold belongs to a set of unique error values E calculated on the learning data set $S_{L}^{\mathrm{AP}}$. For computational reasons, the size of the set E can be reduced, decreasing the precision of its elements.

The optimal threshold is calculated using the formula
(7)$$\begin{array}{cc} t_{p} & ={\mathrm{perc}}_{80}\left\{\sum_{\left(r,t,x,y,z\right)\in S_{L}}\sigma_{t}\left(|{\hat{g}}_{p}\left({\overset{\vee }{r}}_{p}\right)-r_{p}|\right)\;:t\in E\right\}, \end{array}$$
(8)$$\begin{array}{cc} \sigma_{t}\left(x\right) & =\left\{\begin{array}{cc} 0 & \;\mathrm{if}\;x<t\\ 1 & \;\mathrm{if}\;x\geq t. \end{array}\right. \end{array}$$

We classify an AP *p* as one that has changed its characteristics (possibly due to a change of its location) if for an RSS vector r, $|{\hat{g}}_{p}\left({\overset{\vee }{r}}_{p}\right)-r_{p}|>t_{p}$. In such a case, a new localisation model ${\hat{L}}^{p}$ is created, as described in Section 3.1. However, this time the learning data set is $S_{L}^{\mathrm{AP}-\{p\}}.$ In other words, we remove readings from AP *p* from the readings of the learning series $S_{L}^{\mathrm{AP}}$. Therefore, we can define the modified localisation model $m\hat{L}$ by
(9)$$m\hat{L}\left(r,p\right)=\left\{\begin{array}{cc} \hat{L}\left(r\right) & \;\mathrm{if}\;\mathrm{the}\;\mathrm{system}\;\mathrm{does}\;\mathrm{not}\;\mathrm{detect}\;\mathrm{that}\;\mathrm{AP}\;p\;\mathrm{has}\;\mathrm{changed}\;\mathrm{its}\;\mathrm{characteristics},\\ {\hat{L}}^{p}\left(r\right) & \;\mathrm{if}\;\mathrm{the}\;\mathrm{system}\;\mathrm{detects}\;\mathrm{the}\;\mathrm{change}. \end{array}\right.$$

To estimate the quality of the model $m\hat{L}$, it is compared with the system that ideally detects if AP *p* changed its characteristics. Therefore, we introduce $i\hat{L}$, such that
(10)$$i\hat{L}\left(r,p\right)=\left\{\begin{array}{cc} \hat{L}\left(r\right) & \;\mathrm{if}\;\mathrm{AP}\;p\;\mathrm{did}\;\mathrm{not}\;\mathrm{change}\;\mathrm{its}\;\mathrm{characteristics}\;(\mathrm{its}\;\mathrm{location}),\\ {\hat{L}}^{p}\left(r\right) & \;\mathrm{if}\;\mathrm{AP}\;\mathrm{changed}\;\mathrm{its}\;\mathrm{characteristics}\;(\mathrm{its}\;\mathrm{location}). \end{array}\right.$$

The proposed method is universal and works for various localisation models $\hat{L}$. However, as mentioned before, we will work with $\hat{L}=\hat{L}$, i.e., the model that was created using a Random Forest.

## 4. Tests

### 4.1. Simulation of Moving APs

The idea of the tests is to simulate that one of the APs was moved to a new location. This is done by a modification of the learning and testing data sets. First, we choose one AP $p_{0}$ and remove (put aside) its RSS readings from the learning and testing data sets ($S_{L}^{\mathrm{AP}}$, $S_{T}^{\mathrm{AP}}$). Following the above definition, we obtain $S_{L}^{\mathrm{AP}\backslash \{p_{0}\}}$ and $S_{T}^{\mathrm{AP}\backslash \{p_{0}\}}$. Based on such a modified learning data set $S_{L}^{\mathrm{AP}\backslash \{p_{0}\}}$, we create the localisation model ${\hat{L}}^{p_{0}}$, as described in Section 3.1. Then, we choose another AP $p_{1}$ and we replace the readings from that AP in the testing data set $S_{T}^{\mathrm{AP}\backslash \{p_{0}\}}$ with the readings from $p_{0}$. Such a modified testing data set $S_{T}^{\mathrm{AP}\backslash \{p_{0}\}}$ is denoted by $S_{T}^{p_{0}\to p_{1}}$. Next, we calculate the mean, median and gross error for the model ${\hat{L}}^{p_{0}}$ to assess how it works.

It should be stressed that in each case we replace RSS readings with readings received from the same model of AP. Therefore, it is a good but discrete simulation of an actually moving AP. Moreover, the simulation allows us to observe n2=n(n−1)2 unique movements of APs, where is *n* the number of APs in the considered infrastructure.

We prepared several scenarios to illustrate how the changes influence localisation quality and how our system deals with it. The scenarios are sketched in Figure 2.

#### 4.1.1. Scenario 1 (2D Case)

Let us pick a floor f∈{0,1,2,3,4} and two APs $p_{0}$ and $p_{1}$ that are located on this floor. In the first scenario, the 2D localisation model is trained using only APs from a given floor f∈{0,1,2,3,4}, except $p_{1}$, which is put aside. For training, we choose only fingerprints from $S_{L}$ that are located on the floor *f*. Then, in the testing data set $S_{T}$, we choose only fingerprints from floor *f*, and modify the RSS vectors by replacing the RSS reading from AP $p_{0}$ by the readings from AP $p_{1}$. The replacement simulates the location change of one AP, as both $p_{0}$ and $p_{1}$ are the same device models. We do this for every pair $p_{0}$ and $p_{1}$ of APs from floor *f*, obtaining the mean, median and gross errors. Then, to summarise the results, we take the average of mean, median and gross error over all floors f∈{0,1,2,3,4} and all pairs $p_{0}$ and $p_{1}$ of APs from floor *f*. We do not consider the last floor f=5 as the number of APs on this floor is too small (see Section 4.2 for details).

#### 4.1.2. Scenario 2 (2D Case)

The second scenario is similar to Scenario 1, but although we consider only readings from a given floor *f*, the sources of the signals are not limited to APs on this floor. We consider in the localisation process—building the model and locating the terminal—APs from different floors that broadcast on a given floor f∈{0,1,2,3,4,5}. This time, we can include the last floor of the building. Once again, for every floor, we obtain the mean, median and gross errors. Then, we take the average for all three QoS (introduced in Definition 2) measures among all floors *f*, similarly to Scenario 1.

#### 4.1.3. Scenario 3 (3D Case)

In the last scenario—similarly to the other two—we pick two APs, $p_{0}$ and $p_{1}$. However, while training the system, we consider all fingerprints inside the building, without the RSS readings from $p_{1}$. Based on that, we create a 3D localisation model. We test the model on the fingerprints from the testing series, which is modified by replacing the readings from $p_{0}$ with the readings from $p_{1}$. For such a modified testing data set, we obtain the mean, median and gross errors. Finally, we take the average for all three QoS measures among all pairs of $p_{0}$ and $p_{1}$.

### 4.2. Data

The simulations were conducted on data collected in the multi-floor MIS building. Data were collected on six floors in public areas in two independent series. The learning set contains 43,680 fingerprints collected between 18 August 2014 and 22 August 2014. The testing set gathered between 25 August 2014 and 29 August 2014 consists of 46,760 fingerprints. The collected RSS come from a network of 46 identical Cisco Aironet 1140 Series Access Points model AIR-LAP1142N-E-K9. Table 1 presents information about the numbers of APs, fingerprints used in the tests and the tests on each floor. In Scenario 1, the test were not performed on the last floor because of the small number of APs. The fingerprints were included in the tests if and only if they contained at least one AP from the group of APs selected in the given Scenario.

The data were collected using Android OS 2.1 running on mobile phones: HTC One, LG Nexus4 and Sony Xperia. The influence of the phone model on data is discussed in [15]. The location fingerprints were collected in a 1.5 × 1.5 m grid if it was possible, due to environmental limitations. The learning and testing series are 0.75 m apart from each other in each direction. There were 40 fingerprints taken at every measurement point. The measurements were made in four directions parallel to the building axes. We set Ø=−113 [dBm] to note a missing signal.

## 5. Results

This section aims to show the benefits we obtain by using the detection method described in Section 3.2. Therefore, we compare three different systems with an AP *p* being put aside for testing purposes. That is, for a given AP *p*, localisation model $\hat{L}$ we denote by ${\hat{L}}^{p}$ a new model that was created using the same method but using only $S_{L}^{\mathrm{AP}-\{p\}}$ as a learning data set. Thus, we consider three possible modifications. The first, which was already defined, ${\hat{L}}^{p}$, later $m{\hat{L}}^{p}$ and $i{\hat{L}}^{p}$. For a given localisation model $\hat{L}$ and two given APs $p_{0}$ and $p_{1}$, let us introduce the following QoS measure δ that measures, on set $S_{T}^{p_{0}\to p_{1}}$, how replacing readings from AP $p_{1}$ with readings from AP $p_{0}$ increases the mean error.
(11)$$\delta^{p_{0}\to p_{1}}\left(\hat{L}\right)=E(\hat{L},S_{T}^{p_{0}\to p_{1}})-E(\hat{L},S_{T}).$$

Based on the above, we can see the relation between δ and the distance $d=dist(p,p^{\prime })$, by looking at the parametres that describe loss QoS caused by the movement of AP by distance *d*
(12)$$\delta \left(\hat{L},d\right)=\left\{\delta^{p_{0}\to p_{1}}\left({\hat{L}}^{p_{0}}\right):\;dist\left(p_{0},p_{1}\right)\in \left[d-\frac{1}{2},d+\frac{1}{2}\right)\right\},$$
where *d* is an integer.

The results of the proposed system detecting changes as presented in Formula (Equation 9)—labelled as *Proposed model*—are described by the function $\delta (m\hat{L},d)$. In the tests, we compare it with two reference models. The first reference localisation model—labelled as *Not modified model*—is not modified despite the change. The results are described by $\delta (\hat{L},d)$. The second reference localisation model—labelled as *Ideally modified model*—is modified assuming a perfect detection of the change, as presented in Formula (Equation 10). The results are described by $\delta (i\hat{L},d)$.

### 5.1. 2D Scenarios

Figure 3 shows the results for a two-dimensional localisation problem. The results were presented as box plots calculated for sets $\delta (\hat{L},d)$, where d=1 and $\hat{L}\in \left\{\hat{L},i\hat{L},m\hat{L}\right\}$. This allows the reader to compare the error distribution for the proposed and reference models calculated for shifts with one-metre resolution. Figure 3a shows the correlation between an AP shift and the localisation error change that occurs in Scenario 1. Figure 3b shows the same for Scenario 2. We observe that changes performed in a localisation model based on a small number of APs (Scenario 1) result in higher errors than for a model with a larger number of APs (Scenario 2). The maximum error is 15 and 10 m, respectively.

Comparing the results obtained by the proposed system and the ideally modified model—that detects all the changes—we find them very similar. The main difference is the higher standard deviation of errors obtained for very big AP displacements (about 50 m) by the proposed model in Scenario 1. This effect does not occur in Scenario 2.

One can observe that the ideal model and proposed model obtain negative values for shifts less than one metre. This is because the figures show the difference between the localisation error obtained after and before the change. In this case, the elimination of some APs increases the QoS of the localisation system by reducing the localisation error.

Observing the results for the not modified model, we see that the localisation error arises according to the distance. Our system eliminates this tendency and the errors—especially in Scenario 2—are on a similar level for all distances. Moreover, the results obtained by our system are visibly better than for the localisation model without an update. In Scenario 1, the maximum error of the proposed system does not exceed 10 m. The maximum error for the model without the update exceeds 15 m. For Scenario 2, the maximum error is less than 6 m and over 10 m, respectively.

For all compared approaches, the obtained errors are smaller in Scenario 2. This is natural because the number of APs used for localisation on each floor grows six times on average (see Table 1). Therefore, the changes in RSS from a single AP are much less important.

### 5.2. 3D Scenario

Figure 4 shows the results for the three-dimensional localisation case described in Scenario 3. Figure 4a presents the results identically as for the two-dimensional localisation cases, showing the error distribution for the proposed and reference models calculated for shifts with a one-metre resolution. However, because of the discrete vertical shift that could be present in this scenario, we introduced an additional presentation of the error distribution according to the change in the number of floors (Figure 4b).

Analysing the errors according to the distance (Figure 4a) we obtain similar conclusions as for the two-dimensional scenarios. The results obtained by the proposed method are very similar to the ideal model. However, the improvement of QoS observed in the two-dimensional scenarios for a shift smaller than one metre is absent in Scenario 3. The proposed solution improves the results obtained by the not modified model. The maximum error is less than 8 m and over 15 m respectively.

Figure 4b shows that the localisation error grows according to the number of floors between the swapped APs. Once again, the proposed system gives results that are closer to the ideal model than to the model without the update.

For a more formal comparison of the results, statistics for the scenarios were calculated. Table 2 presents statistics calculated for all scenarios. For comparison, the errors obtained in the three-dimensional case are calculated only horizontally.

The reference localisation errors, presented as (*B*), vary from 3.6 to 4.3 m for the mean. After the change (*A*), the mean error rises by 1.2 to 2 m. However, the application of our system (*S*) reduces this growth and the localisation error increases by 0.2 to 0.6 metres only. After the change, the median error grows by 1–1.6 m. Using our system, the QoS drops only by 0.1 to 0.3 m. The main change is visible for gross errors (the 80th percentile). The QoS after the change plunged by 1.7 to 3.2 m. By using our system, the decrease can be reduced to 0.2 to 1 m.

### 5.3. Comparison with other Solutions

We have shown in the previous tests the correlation between error and distance between a previous and future location of an AP. The location change causes a change in RSS. Analysing the collected data, we can say that the RRS change varies between −1 and −73 [dBm]. However, the change can mute the signal entirely and then it is noticed as a missing signal ∅=−113 [dBm]. In such a case, the difference varies between −13 and −86 [dBm].

Therefore, an approach that focuses on missing RSS values to improve localisation results is legitimate. There are two approaches to the missing RSS values: The missing values can be eliminated at the beginning when the radio map is created [12,13,14] or detected dynamically [15].

To compare the two approaches, we created a testbed based on Scenario 1. Because of the limitations of the compared methods, we eliminated from the tests all RSS vectors without ∅ values and consisting of ∅ values only. It was necessary because the method proposed in [15] detects if a terminal should receive a signal from the AP using information from the other APs. For this, the learning set must contain cases describing RSSs from the other APs when the observer AP is turned on as well as when it is turned off. The second method [14] replaces ∅ values with the mean of RSS values from a given AP over the whole learning set. Therefore, vectors consisting of ∅ values only are not allowed.

Using the created testbed, we performed 386 tests. Figure 5 compares the results obtained by the discussed methods using the empirical distribution function. For all methods, δ error was compared. The comparison includes a situation without any preventing methods (No Action), recovering the missing values with the mean (Mean Imputation), detection of missing signals to eliminate their source from the model (List-Wise Deletion) and our changes detection system (Our Solution).

All methods are successful and obtain a better result than ignoring the changes. However, the mean imputation of the missing AP signals has a much higher gross error that the other methods working with the updated model. Among them, the best results are obtained by the system proposed in this work. For our solution and the list-wise deletion model, we observe some negative values. This means, that the updated system gives better results than the system before the change in some cases and the δ error is negative.

Our method obtained a mean error smaller by 1 and 0.5 m in comparison to [14] and [15], respectively. Similarly, the median error is less by 1 and 0.4 m. The difference is the highest for the gross error, where the results obtained by our method are better by 1.8 and 0.8 m. The detailed statistics for all models are given in Table 3.

## 6. Conclusions

We have proposed a dynamic system that detects changes in RSS. The system is based on an estimator that predicts the signal strength of an AP using the readings from all the remaining APs. Using an optimal threshold, the algorithm recognises an AP that has changed its characteristics. Next, the system can rebuild the localisation model excluding the changed AP to keep localisation quality (see Section 3 for details).

We have shown through simulations that the system reduces the error created by a change of AP location by 1 to 1.4 m (see Table 2). Moreover, the system works better than the solutions presented in previous works [14,15] (see Table 3).

The presented system can be used to improve existing localisation systems based on multi-source signals and detect various changes in RSS characteristics. In particular, it can support rank-based fingerprinting and other methods based on infrastructure stability [7,23].

## Figures and Tables

**Figure 1 sensors-20-01828-f001:**
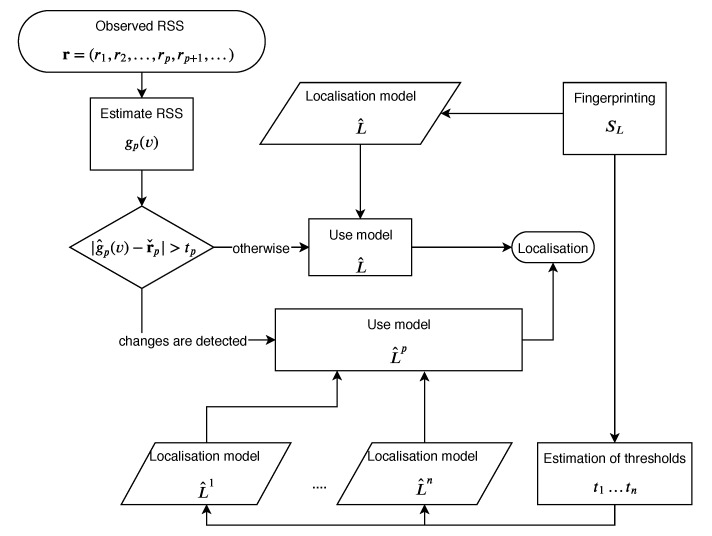
Localisation schema using the characteristics changes detection Indoor Positioning System.

**Figure 2 sensors-20-01828-f002:**
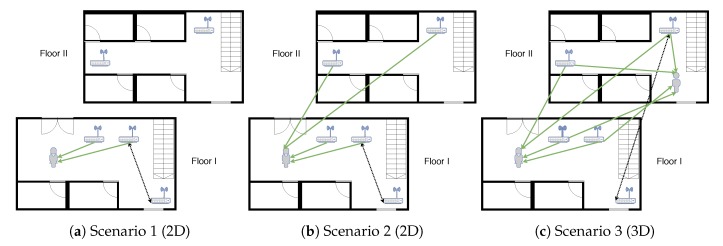
Testing scenarios. (**a**) Localisation using APs from the current floor, horizontal location of APs is changed. (**b**) Localisation using APs from all floors, horizontal location of APs is changed. (**c**) Localisation using APs from all floors, horizontal and vertical location of APs is changed. The green arrows indicate the AP used for the localisation process, the dotted arrow shows an AP shift.

**Figure 3 sensors-20-01828-f003:**
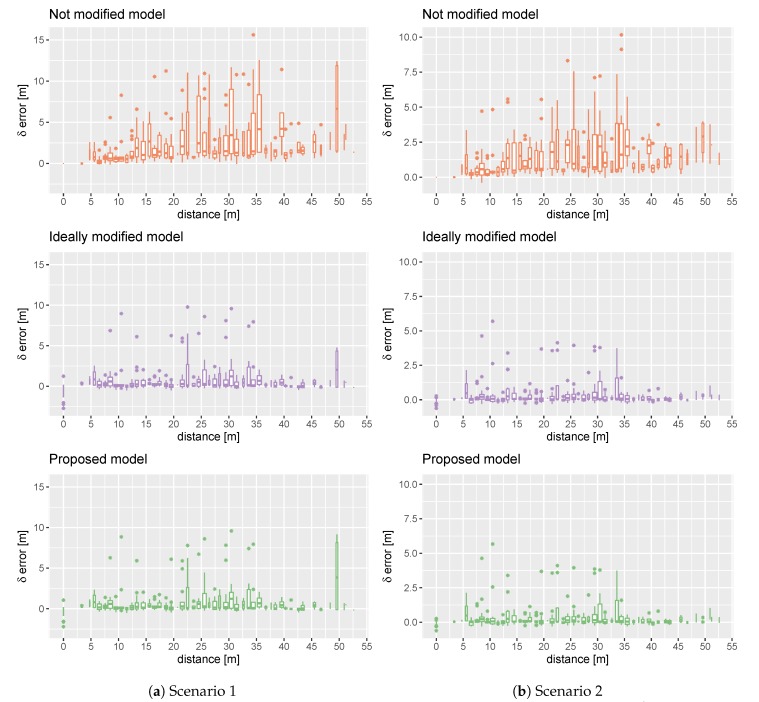
Comparison of error growth according to distance for the proposed model ($m\hat{L}$) and the reference models without modification ($\hat{L}$) and with ideally modified model ($i\hat{L}$) in Scenarios 1 and 2.

**Figure 4 sensors-20-01828-f004:**
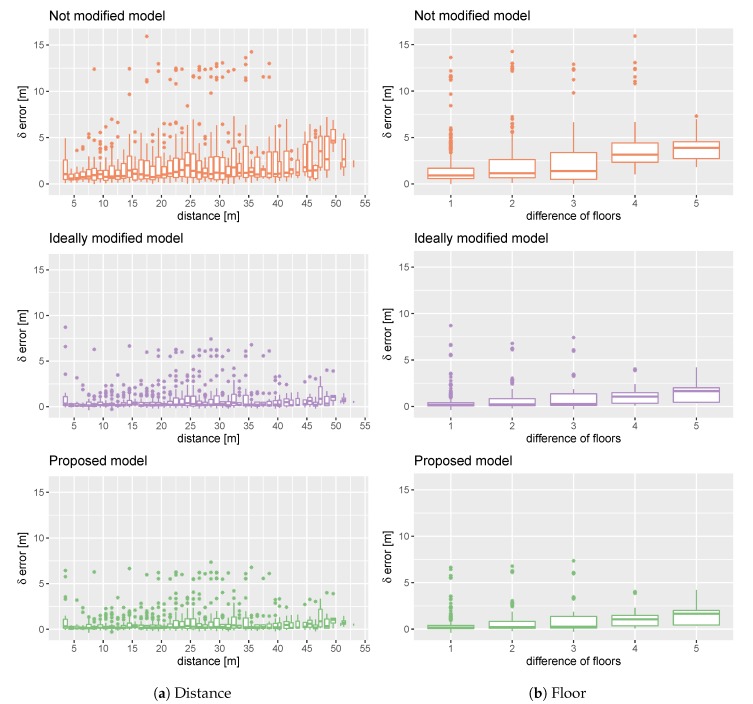
Comparison of error growth according to distance and floor difference for the proposed model ($m\hat{L}$), and the reference models without modification ($\hat{L}$) and with ideally modified model ($i\hat{L}$) in Scenario 3.

**Figure 5 sensors-20-01828-f005:**
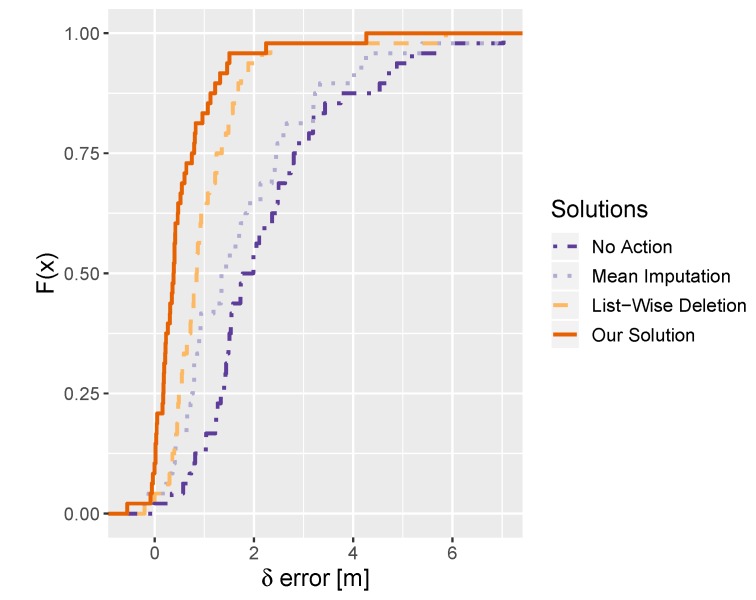
The comparison of the δ errors obtained for various preventing methods.

**Table 1 sensors-20-01828-t001:** Information about floors. The number of the measurement points, the numbers of the present and active APs, the number of the tests for each scenario

	APs			Fingerprints $\left[10^{6}\right]$			Tests		
Floor	I	II	III	I	II	III	I	II	III
0	7	44	44	0.17	0.17	2.30	49	49	259
1	6	38	38	0.26	0.26	1.59	36	36	192
2	13	40	40	1.39	1.39	29.94	169	169	351
3	13	41	41	0.37	0.37	4.05	169	169	364
4	4	36	36	0.05	0.05	1.23	16	16	128
5	3	29	29	0.02	0.02	0.86	0	9	78
∑	46	228	228	2.25	2.25	39.97	439	448	1372

**Table 2 sensors-20-01828-t002:** Comparison of localisation errors before (B) and after (A) the RSS change as well as when the localisation model was rebuilt by our system (S).

	Mean Error [m]			Median Error [m]			Gross Error [m]		
Sc.	B	A	S	B	A	S	B	A	S
I	4.35	6.27	4.91	3.44	5.08	3.78	6.39	9.54	7.28
II	3.56	4.77	3.81	2.76	3.73	2.84	5.19	6.92	5.42
III	3.74	5.72	4.35	3.00	4.41	3.20	5.18	8.43	6.18

**Table 3 sensors-20-01828-t003:** Comparison of localisation errors for various preventing methods

Solution	Mean Error [m]	Median Error [m]	Gross Error [m]
No Action	2.07	1.79	3.39
Mean Imputation [14]	1.57	1.40	2.74
List-Wise Deletion [15]	1.05	0.77	1.74
Our solution	0.59	0.38	0.93

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
