# Peer review of "Automatic Detection of Changes in Signal Strength Characteristics in a Wi-Fi Network for an Indoor Localisation System"

_sensors, 2020, doi:10.3390/s20071828_

Round 1

Reviewer 1 Report

To improve the readability of the paper, the abstract and (most important) Section I should briefly introduce the proposed solution by anticipating few technical details of the algorithm. While of course it is not possible to fully present the proposed solution in Section 1, by anticipating its most crucial aspects can improve the interest of a reader. Moreover, even the Conclusions section should very briefly outline the proposed solution, not only providing a reference to Section 3.

Section 2 should clearly point out how the proposed solution differs from and improves state-of-the-art literature. Otherwise, readers are not able to appreciate the novelty and the potential impact of the proposed solution.

The paper states that presented performance results are based on simulations. However, authors present a real-world environment thus letting readers to think that performance are based on an actual environment exploited to gather RSS values. Then rereading the paper, it seems that RSS values are gather in a real-world scenario but then the movement of an AP is simulated by artificially modifying some RSS values. Authors should clarify this aspect. I also encourage authors in presenting performance results based on a real-world scenario by actually moving an AP.

Finally, Fig. 3, 4, and 5 should be clarified, since there are many lines and colored areas making difficult its comprehension. I also suggest to provide labels more expressive than “ideally modified” and “System modified”. Similar considerations apply for Fig. 6.

Other comments:

 - Abstract: extend the sentence “It can be applied to any service based on Wi-Fi network to various kinds of changes.” by adding some examples

 - Abstract: I suggest “from other locations” or “from other localization procedures” I place of “from other localizations”

 - I would suggest to reduce the amount of “new lines” by grouping multiple sentences in the same paragraph

 - Section 1: revise the sentence starting with “Such the sources”

 - Section 1: “we propose a detection algorithm” in place of “we propose the detection algorithm”

 - Section 1: revise the sentence starting with “Because of the non-artificial” to make easier its comprehension

 - Section 1: I suggest “our solution reduces” in place of “we could reduce”

Reviewer 2 Report

This paper propse a RSS based indoor positioning system to improve the Wifi service quality. My concern is as follows.

1, There are many typos in the paper. For example, in the introduction section:

1), are more and more commonly. ---- common

2),  and require a specific level ---- requires

3), by comparing an RSS vector ---- a

4), and sensor technology, have a significant ---- remove the comma

5), have a significant impact on Quality of Service (QoS) ---- QoS is already defined in the abs

There are numerous other errors in the entire paper. The authors should locate and correct all of them in the revision.

2, This paper lacks a specific description of its contributions, and hence, the main novelty. 

3,I think the key of the algorithm is to derive the targetting AP location, i.e., rp, from the readings of the other APs. However, in the 3. Methodology section, the algorithms details are missing. It is very hard to judge the quality of the paper with such limited descriptions.

4, Even if the method is solid and novel, the authors still need to consider other real-world factors. For example, what about noises? Using only one vector to determine the long term change of the APs should make it very vulnerable to various noises. Moreover, if the changes are caused by ageing, there should be multiple Aps changing their characteristics, while the paper only considers the case of one AP changing.

Reviewer 3 Report

The manuscript deals with the indoor localization model based on fingerprinting of RSS Wi-Fi readings. The problem of the model vulnerability to different major changes in the infrastructure is considered such as Access Points relocation. While many papers propose different solutions for indoor localization based on Wi-Fi, very few of them consider the problem of changing infrastructure. This manuscript seems to be a continuation of the previous papers of both authors, where they consider the influence of the changes of the Wi-Fi infrastructure on the accuracy of a localisation model and propose a solution of automatic detection of missing Access Points. This paper considers a general case of detection not only missing APs, but also those which changed their characteristics that may influence the accuracy of the localisation. It looks like their approach is different from the previous ones. Although their method is tested for a particular localisation model it looks like it can be applied to any localisation model that was created using RSS fingerprinting method.

Authors obtained substantial improvements in accuracy between corrected and not corrected model. However, one of my concerns is that based on every RSS reading (so almost every second) their model checks if a given AP changed its characteristics and based on that it changes the localization model. It looks that it can be too frequent and may be computationally very demanding if one considers the environment with many users and APs.

I have following comments to the manuscript:

- localisation or localization, the usage of localization is more often in the community.

- state of art – state of the art – “the” is missing

- all acronyms in the text should be explained in the text after the first use, e.g. IPS, CRIL etc.

- Figure 6 – there is probably mistake, because there is negative error achieved by some systems. What does it mean? Can we obtain error less than 0 m?

Round 2

Reviewer 1 Report

In Section 2, I suggest "Designing our solution" in place of "Preparing our solution", "the designed" in place of "the prepared", and "to design" in place "to prepare".